# OpenReview forum: "How Reasoning Evolves from Post-Training Data: An Empirical Study Using Chess"
_ICML.cc/2026/Conference — ICML 2026 regular_

### Official Review · Reviewer_Mkjs · 2026-03-06

**Soundness:** 3
**Presentation:** 2
**Significance:** 3
**Originality:** 3
**Overall Recommendation:** 4
**Confidence:** 3

**Summary:**

This paper explores how reasoning capabilities in LLMs evolve during the transition from SFT to RL. Using chess as a verifiable testbed, the authors analyze how different types of training data influence model performance, reasoning faithfulness, and hallucination rates. Specifically, they find that data with multiple tasks, data dense with high-quality tokens, diverse data, and data tht encourage long-horizon thinking elicit better reasoning.

**Compliance With Llm Reviewing Policy:**

Affirmed.

**Final Justification:**

My concerns are mostly addressed by the authors. I will maintain my positive scores toward acceptance.

**Key Questions For Authors:**

1. What is the difference between "multitask training" and "dataset diversity" in section 4.1? Isn't multitask training inherently diverse in dataset?
2. It seems that Best Move - All Data achieves the best performance overall. Why do the authors claim that Best Line is better than Best Move?

**Limitations:**

Yes

**Strengths And Weaknesses:**

Strength:
- It is quite a novel idea to use chess to isolate the data contamination problem common across model LLMs accessible today
- Via detailed ablation studies, the authors successfully identify several factors that lead to better reasoning in post-training, including multi-task data, diverse data, and density of high-quality tokens, inspiring the development of more efficient post-training data pipelines.

Weakness:
- The organization of the passage is a bit messy, perhaps because the authors are trying to put together too many results. It seems that the effects of the dataset on training quality for SFT and RL are sometimes mixed (eg, in Section 4.1). However, a rigorous analysis should separate the effect between SFT and RL, respectively, as they are inherently different training paradigms.
- The conclusion that "most effective datasets were dense with difficult, high-quality tokens" is not rigorous enough. An alternative and plausible explanation for the lower performance trained with factual board answering could be a greater question type mismatch between the training and eval.

---

> ### Author Rebuttal · Authors · 2026-03-31
>
> Appreciate the feedback!
>
> First, we wanted to point out (see bottom of this comment) that we've completed an additional experiment we plan to include in the final version. This experiment quantifies 'token information density' -- we train a small language model (Qwen2.5 0.5B Instruct) on a fixed token budget from each of our datasets and can see what portion of the tokens become trivial.
>
> For example, on Factual Board Answering it seems it's a bit 'too easy' -- after training on 8mm FBA tokens (4mm / epoch @ 2 epochs), the model assigns a probability >0.995 to the correct token on 51.90% of our validation set. Or the VABP data -- it becomes 'memorizable' with the SFT'd small model assigning a probability >0.995 to 70.92% of tokens in our validation set.
>
> Compare these to the Best Move or Best Line data, we clearly see the datasets are much more 'information dense' as predicting those tokens remains more non-trivial. We hope you agree this improvement is beneficial and adds rigor.
>
> Regarding organization and combining SFT + RL, we understand this concern; we went through several iterations writing this paper to try to get this right given lots of different results and page limits. But we would push back regarding splitting out SFT + RL -- we chose to control for the RL stage (same algorithm, same tasks, same number of samples across cohorts) so that we can understand how changes at the SFT stage influences downstream performance. Since our intent was to study 'SFT data -> reasoning behavior' and our choice was to keep the RL stage consistent, we felt the best way to display results + discuss takeaways is w.r.t. the data trained on (SFT stage) hence why SFT seems combined with RL.
>
> And totally agreed SFT and RL are different paradigms. That's also why nearly all results / analyses include SFT and RL side-by-side so that the reader can understand how the RL step evolves the SFT checkpoint. This was our way of trying to toe the line of 'our experimental variable is SFT data but we study downstream performance' while also showing how RL impacts performance.
>
> Please let us know if you have any further questions / if you have any recommendations on how to improve layout. Happy to consider alternatives.
>
> ---
>
> ## Information Density Experiment
>
> We took each of our 6 datasets (Best Line, Best Move, Factual Board Answering, Guided Synthetic, Verbalized Alpha-Beta Pruning, and Rejection Sampling) and trained the Qwen2.5 0.5B language model via SFT on 4mm tokens from each dataset separately (for 2 epochs, so total of 8mm tokens). The intent is to then study how 'trivial' the datasets are by analyzing the logit probabilities assigned to each token on a held-out validation set -- this token-level predictability serves as a proxy for the conditional entropy (under next-token prediction) of each dataset.
>
> We can compare predictability of a held-out validation set of tokens under the Pre-SFT (Qwen2.5 0.5B Instruct) and Post-SFT models:
>
> | Dataset | Correct Validation Tokens > 0.995 |  | Correct Validation Tokens < 0.5 |  |
> |--------|--------------|--|-------------|--|
> |        | Pre-SFT      | Post-SFT | Pre-SFT | Post-SFT |
> | Best Move | 0.05% | 28.83% | 96.77% | 36.70% |
> | Best Line | 0.01% | 24.07% | 87.27% | 46.42% |
> | Factual Board Answering | 0.09% | 51.90% | 94.68% | 19.38% |
> | Guided Synthetic | 2.38% | 9.19% | 83.13% | 52.80% |
> | Rejection Sampling | 12.50% | 20.94% | 63.14% | 44.37% |
> | Verbalized Alpha-Beta Pruning | 3.78% | 70.92% | 57.00% | 17.47% |
>
> Now, raw token predictability is insufficient on its own. This does show us that some datasets are more easily 'learnable' -- whether that be memorization (in VABP) or possibly task difficulty (per FBA) -- this is insufficient on its own. A dataset can have seemingly high-conditional entropy if it is noisy or weakly constrained. What we also care about is the task-relevant information-density -- in our case, how much 'chess information' is embedded in each dataset.
>
> Therefore, we must also consider the density of chess-play relevant tokens in each dataset. Best Move and Best Line have high density of chess-play tokens -- the model is tasked with predicting the precise move or line Stockfish would play. Other datasets, such as Rejection Sampling or Guided Synthetic, have more natural language included given they are synthetic -- this synthetic natural language explains the limited token-level predictability. Thus while the synthetic datasets are less effective at eliciting strong chess play on their own (despite remaining less predictable through SFT), we find they provide a necessary prior for reasoning behavior that is synergetic when combined with the dense, difficult tokens of Best Move and Best Line.
>
> We intend to include additional visualizations (probability of tokens in validations samples) and other experimental results that compare Best Move and Best Line entropy in the appendix as well.

---

> > ### Author Rebuttal · Reviewer_Mkjs · 2026-04-02
> >
> > Thanks for the response. I think the table on information density helps improve the overall rigor. I suggest the authors include one more column in the table outlining the corresponding model performance improvement when trained on that dataset, so that readers can more intuitively compare.
> >
> > When the authors mention "controlling for RL stage", do the authors mean that the RL stage uses the same data composition as the respective SFT stage, or all all experiments use the same RL data composition but with different data composition during SFT?

---

> > > ### Author Response · Authors · 2026-04-03
> > >
> > > Will consider adding!
> > >
> > > When we say 'controlling for the RL stage' -- we mean that we hold the following the same during the RL stage across all runs:
> > > - Evaluation samples used (# of samples, samples per eval task; we use the same samples as well)
> > > - Hyperparameters (batch size, learning rate, num rollouts per sample for GRPO, algorithm)
> > >
> > > So same RL data composition yes, the only thing that changes across experiments is the SFT checkpoint (which is directly influenced by the data we train on during SFT).

---

### Official Review · Reviewer_FsFd · 2026-03-08

**Soundness:** 2
**Presentation:** 3
**Significance:** 3
**Originality:** 3
**Overall Recommendation:** 4
**Confidence:** 4

**Summary:**

This study investigates how post-training (SFT&RL) influence the evolution of reasoning capabilities in language models within the chess environment. Using the Qwen2.5 7B-Instruct model, the authors compare the performance of various datasets across SFT and RL stages, leveraging chess engines to provide verifiable rewards. The authors derive several observations, including how different dataset constructions affect SFT and RL performance, and which metrics can predict final RL outcomes.

**Compliance With Llm Reviewing Policy:**

Affirmed.

**Final Justification:**

Based on the rebuttal, although restricting experiments to one model family limits the broader applicability of the conclusions, I recognize that this work offers sufficient value to the community. Therefore, I update my overall recommendation to Weak Accept.

**Key Questions For Authors:**

While the empirical results are solid and comprehensive, to what extent do you think these conclusions generalize beyond the specific setting? Can insights derived from Qwen2.5 7B and your particular RL configuration be extended to wider RL and SFT methodologies? Thanks.

**Limitations:**

yes

**Strengths And Weaknesses:**

Strengths:

The roles and interplay of SFT and RL in post-training is an important topic in LLM training research. By using the chess environment as a testbed, this paper studies the impact of SFT and RL on training reasoning models, and provides comprehensive evaluation metrics within this domain—including move legality, quality rankings, hallucinations, reasoning strategy, reasoning qualities—thereby going beyond single accuracy metrics. Overall, the experimental design and results are solid.

Weaknesses:

The paper generally lacks technical depth. While the experiments are comprehensive, most results remain at the level of score reporting without rigorous theoretical analysis or mechanistic interpretation. Furthermore, factors such as the choice of RL algorithms and base models may significantly affect the experimental results and observations. Consequently, although the findings offer insights specific to the chess domain, the conclusions may be difficult to generalize to broader contexts of LLM training.

---

> ### Author Rebuttal · Authors · 2026-03-31
>
> Thanks for the review! We acknowledge your criticisms but want to offer a perspective and address your question re: takeaway generalization.
>
> Our goal was to explicitly study 'reasoning' -- specifically, what the model 'verbally' outputs -- and, more generally, start to prod at this difficult question of 'What training data gives you a good checkpoint for reasoning post-training?'
>
> Now there are several great existing papers we knew of that we didn't want to recreate. For example, mechanistic interpretability on Othello (Nanda, Lee et. al., 2023 -- "Emergent Linear ...") and 'Grandmaster-Level Chess without Search' (Ruoss et. al., 2024). Techniques from the former we felt were out of scope and marginally informative given our other methods, the latter is why we focus on trying to improve chess ability via reasoning / study reasoning behavior (i.e., if we just wanted strong chess performance, the Ruoss paper showed you can essentially distill a chess engine into a 300M parameter transformer. So just lob data + compute and you have a grandmaster!)
>
> Rather, we wanted our contribution to be different -- hence our angle of studying reasoning behavior (both quantitative and qualitative performance). Most papers focus on domains like math or code -- but these have been *heavily pre-trained for* -- so what amount / type of data would it take to get reasoning to work in a novel domain that the model isn't great at? This was our intent.
>
> We believe our contributions are meaningful for the following reasons:
> - We study how multiple principled datasets influence performance and downstream reasoning (through SFT and RL) -- and on each (all discussed in Section 4), we study hallucination rates (Appendix E), reasoning strategy usage (Appendix F), and reasoning quality via LLM-judge (Appendix G). We study these from Baseline -> SFT -> RL and we feel this is quite a holistic analysis of verbal reasoning behavior. This breadth and depth of study across datasets in a fairly controlled domain is not something we have seen before -- given many people understand chess well, we feel these takeaways can be easily understood and help researchers think about their own dataset curation and be aware of pitfalls (e.g., unfaithful reasoning)
> - We plan to include an additional experiment (posted under Reviewer Mkjs) analyzing information density of each dataset which we believe further helps researchers understand how to think about data curation in the reasoning paradigm. We feel this adds technical depth and further contributes to answering the question of 'how data influences reasoning' -- this can help folks think about how they design their own data
> - We also show that there are several strong statistical signals you can use to predict downstream post-RL performance from an SFT checkpoint (e.g., 'reasoning quality' as judged by an LLM). While we only study chess, we show this across many different datasets -- we would hope a researcher thinking about midtraining recipes considers using these methods in their own midtraining evaluation
> - We are also open-sourcing all datasets, large-scale model checkpoints, and various data generation / analysis code for people who may be interested in further pursuing LLM chess reasoning models
>
> So our choice of a targeted domain (chess) felt necessary in order to do the controlled studies we sought in order to answer our desired question (SFT data -> reasoning behavior). However, we believe many contributions extend beyond chess -- if you're a researcher trying to better understand how data influences reasoning behavior, we feel there are a lot of takeaways you can intuit and apply to your own work from our paper given we attack this from many angles (quantitative performance, qualitative performance, predictability of downstream performance, general dataset design).
>
> Beyond this, we wholly agree that using a single model family / RL algorithm is a limitation (we mention this exact thing in our 'Limitations & further discussion') -- we are forced to operate within compute constraints.
>
> Would love to answer any further questions regarding your hesitations!

---

> > ### Author Rebuttal · Reviewer_FsFd · 2026-04-03
> >
> > The rebuttal is clear. The authors clearly describe the insights and logic behind their experiments. Apart from the unresolved issue of using only a single model family / RL algorithm, I think the authors have addressed my other concerns. This paper investigates post-training for reasoning using chess; however, I still question whether experiments based solely on Qwen are sufficient.

---

### Official Review · Reviewer_3E9r · 2026-03-12

**Soundness:** 3
**Presentation:** 2
**Significance:** 2
**Originality:** 3
**Overall Recommendation:** 4
**Confidence:** 3

**Summary:**

This work investigates how to develop reasoning capabilities in language models for chess. The authors choose chess as a testbed because LLMs have struggled with chess. In addition, it offers several convenient properties like established theory, clear episodic MDP structure, large-scale data availability, and  an efficient oracle (like chess engines) that provides verifiable rewards and enables high-quality synthetic data generation. The authors leverage chess engines to synthesize several types of training data which includes best move, best Line, verbalized alpha-beta pruning, rejection sampling. There are several key findings of the paper.  1. Multitask training proves beneficial which yields higher move quality, reduced reward hacking, and more robust models compared to single-task training. The combination of Best Move and Best Line is the optimal choice.  2. Models trained on Best move data exhibit unfaithful reasoning after RL. Their reasoning traces become disconnected from their final move choices. On the contrary, models trained on Best Line data largely retain faithful reasoning through RL.  3. The authors identify several SFT-stage indicators that predict downstream RL performance through linear regression analysis.  The referenced move accuracy and reasoning quality are statistically significant predictors. Empirically 7B-parameter model surpassing the leading reasoning models including gpt-oss-120b.

**Compliance With Llm Reviewing Policy:**

Affirmed.

**Final Justification:**

The authors have  resolved my concerns raised in the rebuttal. I find the exploration of reasoning capabilities in language models for chess to be a compelling idea. I will keep my rating at 4.

**Key Questions For Authors:**

See my comments on weakness.

**Limitations:**

yes

**Strengths And Weaknesses:**

Strength

The work consider an  interesting problem. Chess data is rarely emphasized in pretraining, making it an ideal controlled setting to analyze which data types are effective for SFT and RL without confounding factors from prior exposure.


The finding regarding faithful versus unfaithful reasoning offers valuable insight. In particular, the observation that multi-step trajectory data yields faithful reasoning is intuitively compelling, as it corresponds to multi-step rollouts of the Bellman equation that explicitly encode transition dynamics and value estimation.


The use of programmatic generation with verifiable rewards ensures data quality and scalability.



Weakness

The experiments are confined to a single model family, Qwen2.5 7B-Instruct. It remains unclear whether the conclusions about data efficacy, RL stability, and faithfulness generalize to other architectures. I suggest the author replicate their experiments on at least one additional model family to validate the robustness of their findings.

The evaluation focuses heavily on move quality metrics rather than holistic game play.


The paper notes that Verbalized Alpha-Beta Pruning underperforms despite being hallucination-free and incorporating explicit search structure. Can the authors explain more why it is so ？

---

> ### Author Rebuttal · Authors · 2026-03-31
>
> Thank you for the review!
>
> Wanted to address some of the weaknesses you mention:
> - We definitely agree that it would have been beneficial to extend this work to more model families (beyond Qwen2.5) -- we explicitly mention that in the 'Limitations & further discussion' section. Due to constraints this wasn't pursued
> - Regarding single-move vs. multi-turn (i.e., full game play), we also agree this is a very promising direction. We similarly discuss this in the 'Limitations & further discussion' section as we believe multi-turn RL could mitigate some of the reward hacking behavior you experience when focusing on the single-turn setting (which we focused on). Fully agree this is a promising direction, but we felt this warrants future work and is beyond the scope of this specific study (we highlight it as a future study)
>
> Regarding why Verbalized Alpha-Beta Pruning underperforms, we actually just completed an information density analysis on each of our datasets that we plan to include in a final version (more detail posted under Reviewer Mkjs). The gist is that we take a small language model (Qwen2.5 0.5B Instruct) and SFT it on ~10mm tokens from each dataset. We then measure probabilities assigned to each token in a validation set as a proxy for 'conditional entropy'.
>
> What you find with the Verbalized Alpha-Beta Pruning dataset is that prior to any SFT, the Qwen2.5 0.5B Instruct model chooses the correct next token with probability >99.5% on only 3.78% of the tokens. After 2 epochs of training (4mm tokens per epoch of only VABP data), this increases to 70.92% of tokens. Essentially, much of the tokens are 'memorizable' -- we have a pretty neat visualization of a sample in the dataset where you can see exactly which tokens still retain uncertainty -- many are where our programmatic generation has stochastic choice between phrases, some of which is when an influential 'chess pivot token' is chosen.
>
> We mention in our paper that we believed VABP had many memorizable tokens -- this analysis that we plan to include in the final version validates / solidifies that hypothesis.

---

> > ### Author Rebuttal · Reviewer_3E9r · 2026-04-03
> >
> > Thanks for the reply. I will keep my positive score.

---

### Official Review · Reviewer_pXpy · 2026-03-13

**Soundness:** 3
**Presentation:** 2
**Significance:** 3
**Originality:** 3
**Overall Recommendation:** 5
**Confidence:** 4

**Summary:**

The authors show that you can train an LLM to play chess better by using RL. They are particularly interested in how the training data and training task effect the model's performance. There work suggests that higher performance can be achieved by a strong target and diverse dataset

**Compliance With Llm Reviewing Policy:**

Affirmed.

**Key Questions For Authors:**

Please label your plot axis. What is the point of figure 4 if the x-axis is undefined, and fig 2 could be a table.

Did you asses if the models can do other chess related tasks after training? i.e., does this result in any type of generalization?

**Limitations:**

Why didn't you have the models play a tournament as is standard when creating a chess engine (best move predictor)?

**Strengths And Weaknesses:**

# Soundness

This paper struggles with the same soundness issues that other LLM papers suffer from in the space, specifically that chess evaluation tend to be binary, so constructing more complex assessments is fraught. The faithfulness measures I am doubtful of as they rely on a model which sufferers from the same issues they are seeking to alleviate. That said as that is not the main result.

As far as I can tell the experiments were only conducted once, and the authors never assessed the actual playing performance of the models. This limits the strength of the their claims

# Presentation

The paper is clear, but most of the figures are bad. If you're only presenting a few numbers why aren't you using a table? That aside I  think the paper can meet the bar for ICML with minor improvements

# Significance

This appears to be a step forward in creating a language model that can play chess without needing significant assistance, but they have not fully cracked it. So this is of moderate significance

# Originality

The authors claim that their assessment of learning over time is novel, the other components are less novel so this again has moderate significance

---

> ### Author Rebuttal · Authors · 2026-03-31
>
> Appreciate your comments! Couple quick responses:
> - Re: Fig. 4 and Fig. 2 -- agreed with everything you mention. We'll make these changes
> - You ask if we tested generalization to other domains (beyond chess) -- we didn't pursue this because our intention was to focus on how data influences reasoning in a controlled setting. We felt testing broader generalization was out of scope and less relevant to our target problem
>
> Additionally, you ask 'Why didn't you have the models play a tournament'. Very valid!
> - We primarily focused on the single-turn setting which is evalauted using the 'Predict Move' task we created. That eval tests 'given random board, play a move' -> score based on how highly Stockfish ranks that move
> - Now we fully agree that the multi-turn setting (full game-play) is interesting and discuss this in the 'Limitations & further discussion' section. Multi-turn RL brings more complexity (especially around compute costs and infra) -- we feel this is beyond the scope of the current paper and is why we discuss this as a future direction. We personally believe the multi-turn setting would actually mitigate some of the reward hacking behavior we saw in the single-turn domain -- it would also lead to more interesting 'full-game play' analysis. Definitely promising for follow-on work, but beyond scope
> - We'd also like to quickly note for context -- language models *really* struggle with full-game chess play. Our base model (Qwen2.5 7B) could only play a legal move 8% of the time -- this is not sufficient to get baseline gameplay performance. This inability to play legal moves extends well beyond Qwen2.5 as well -- we reference the 2025 Kaggle AI Exhibition where models such as Kimi2 (1T params but non-thinking variant) lost all four of its first round games because it couldn't play legal moves; this 'loss by repeated illegal moves played' also occurred with other near-frontier models such as Gemini 2.5 Flash and DeepSeek-R1. *Language models really struggle with chess.*
>
> Also wanted to mention we completed a 'Token information density' experiment that we plan to include in the final version (summary posted under Reviewer Mkjs).

---

> > ### Author Rebuttal · Reviewer_pXpy · 2026-04-04
> >
> > Thank you for your response.
> >
> > To clarify, I asked if you tested other tasks after training, not non-chess tasks. I'm asking if you believe this task will generalize.
> >
> > Your responses about why you did not have the model play games don't match the literature. LLMs can play chess fine, Allie[1] is based on GPT-2 and is available to play now[2]. You just need to mask illegal moves from the policy outputs like you would with other types of structured output.
> >
> > [1]https://arxiv.org/abs/2410.03893
> > [2] https://lichess.org/@/AllieTheChessBot

---

> > > ### Author Response · Authors · 2026-04-05
> > >
> > > Apologies for misunderstanding.
> > >
> > > Regarding generalization to other chess tasks, we do have some experiments in Appendix D -- see Table 3. What we show in this table is the following:
> > >
> > > - **Performance on general Factual Board Answering (FBA) questions**. This asks various board understanding questions such as 'How many spaces can the white rook on a6 move to?' or 'Compute the material advantage given a queen is worth ...'. There is a bit more noise here because some of the models are trained on similar FBA tasks (which we note with footnote 'c') and for those that are, we see instruction following errors (sometimes the model immediately responds with the answer instead of using answer tags given the direct response was the SFT data format...no answer tags would be counted incorrect).
> > > - **Performance on Out-of-Distribution Mates**. These are sourced from a recent paper (Mészáros et al., 2025) and we have a few samples in Figure 18. The majority are situations our trained model would have never seen in training data given they are illegal / impossible situations.
> > >   - Candidly, we hoped our model would have performed better vs. gpt-oss-120b; however, it is worth flagging that our baseline score is 0% (for Qwen2.5 7B Instruct) and both SFT and RL push performance in excess of a model >50x the size in parameter count (Llama 4 Maverick).
> > >   - This may be more a testament to the generalization reasoning ability of gpt-oss-120b or that gpt-oss-120b may have been trained on 'mate puzzles' as an RL environment. We don't know their mix.
> > >   - It may also highlight a reward hacking behavior we found from our models which favor 'conservative play' -- we discuss in the paper how multi-turn RL or training on chess puzzles (specifically, multi-turn chess puzzles) would be a promising direction to counteract this reward hacking
> > >
> > > **So a general takeaway is we do see some chess generalization**. That said, our goal was to study how reasoning evolves from data -- not necessarily to build the best chess bot -- and we do note methods we believe could push chess performance.
> > >
> > >
> > > Now as to LLM chess play -- we disagree that LLMs can play chess. Couple thoughts:
> > >
> > > **Allie isn't really a language model**. They take GPT-2 weights but use a new vocabulary ~2000 tokens long that represents each possible move in UCI notation. It's not token masking. They then train this GPT-2 transformer to predict the next move given previous moves -- they train this on ~250B chess move tokens and at no point use language as they only use their new 2k token vocab. Allie as input uses a few 'meta' tokens followed by earlier moves -- then it outputs a distribution over next moves (they also have two other heads, one predicts 'win probability' and one predicts 'move time'). Their usage of the word 'ponder' is a bit stretched as they simply use 'move time' in one of their algorithms to set MCTS search depth.
> > >
> > > Now the Allie approach is similar to the Ruoss paper we discuss -- the Ruoss paper showed that you can train a transformer architecture (in Ruoss, without any search / MCTS) to reach grandmaster-level (they test 'Behavior Cloning' which is most similar to Allie and show this gives strong results). Both of these works, in our opinion, are an endorsement of the transformer architecture + the power of deep learning on large datasets. We also discuss GPT-3.5-Turbo Instruct which essentially shows this as well -- no 'language-guided reasoning' on chess but strong performance if you train it specifically on chess gameplay.
> > >
> > > So a few final points on LLM chess play:
> > > - Current frontier / near-frontier models still struggle to reason with language to play effective chess
> > >   - As we noted -- we see simple blunders + playing legal moves. Some are better than others (e.g., some chess experts estimated OpenAI o3 at ~1500 Elo after the Kaggle comp), but given 'superhuman' ability in math and coding, some may feel this is a bit disappointing w.r.t. skill generalization
> > > - Ruoss et. al., Allie, others have shown you can distill strong chess play into a transformer (with ample data + parameters) -- but these don't use language-based reasoning. The promise of reasoning is that you can get strong generalization with relatively small amounts of SFT + RL data -- distilling Stockfish with ~1T tokens isn't necessarily aligned with that
> > >
> > > Lastly, reminder that our paper was to study language-based reasoning and how your training data influences reasoning behavior -- not necessarily produce the strongest chess model.
> > >
> > > But LLM chess ability is indeed an interesting area to watch! Chess is obviously well-suited for RL and we expect labs will train on more chess data as RL is scaled further and classic verifiable domains are further exhausted.

---

### Decision · Program_Chairs · 2026-04-30

**Decision:**

Accept (regular)

**Comment:**

This paper investigates the development of reasoning capabilities in Large Language Models (LLMs) using chess as a controlled, verifiable testbed. By isolating the impact of different Supervised Fine-Tuning (SFT) data compositions on downstream Reinforcement Learning (RL) performance, the authors provide insights into how specific data types influence model reasoning, faithfulness, and hallucination rates.

The main Weaknesses & Rebuttals includes,

- Scope of Generalization: focusing on chess and Qwen models

- LLM Chess Capabilities

All reviewers recommend acceptance. Despite limitations in model variety, the depth of the data analysis and the clarity of the results regarding reasoning faithfulness provide sufficient contribution to the ICML community.